# Microglial Metamorphosis in Three Dimensions in Virus Limbic Encephalitis: An Unbiased Pictorial Representation Based on a Stereological Sampling Approach of Surveillant and Reactive Microglia

**DOI:** 10.3390/brainsci11081009

**Published:** 2021-07-30

**Authors:** Leonardo Sávio da Silva Creão, João Bento Torres Neto, Camila Mendes de Lima, Renata Rodrigues dos Reis, Aline Andrade de Sousa, Zaire Alves dos Santos, José Antonio Picanço Diniz, Daniel Guerreiro Diniz, Cristovam Wanderley Picanço Diniz

**Affiliations:** 1Núcleo de Pesquisas em Oncologia, Programa de Pós-Graduação em Oncologia e Ciências Médicas, Hospital Universitário João de Barros Barreto, Universidade Federal do Pará, Belém 66073-005, Brazil; leonardocreao@gmail.com (L.S.d.S.C.); cwpdiniz@gmail.com (C.W.P.D.); 2Laboratório de Investigações em Neurodegeneração e Infecção, Hospital Universitário João de Barros Barreto, Instituto de Ciências Biológicas, Universidade Federal do Pará, Belém 66073-005, Brazil; bentotorres@gmail.com (J.B.T.N.); camilamendesdelima@gmail.com (C.M.d.L.); rrreis88@gmail.com (R.R.d.R.); alinebio02@yahoo.com.br (A.A.d.S.); zaire.santos@gmail.com (Z.A.d.S.); 3Faculdade de Fisioterapia e Terapia Ocupacional, Universidade Federal do Pará, Belém 66075-110, Brazil; 4Laboratório de Microscopia Eletrônica, Instituto Evandro Chagas, Belém 66093-020, Brazil; joseantonio@iec.gov.br

**Keywords:** viral encephalitis, microglia, quantitative neuropathology, piry virus, albino swiss mice

## Abstract

Microglia influence pathological progression in neurological diseases, reacting to insults by expressing multiple morphofunctional phenotypes. However, the complete morphological spectrum of reactive microglia, as revealed by three-dimensional microscopic reconstruction, has not been detailed in virus limbic encephalitis. Here, using an anatomical series of brain sections, we expanded on an earlier Piry arbovirus encephalitis study to include CA1/CA2 and assessed the morphological response of homeostatic and reactive microglia at eight days post-infection. Hierarchical cluster and linear discriminant function analyses of multimodal morphometric features distinguished microglial morphology between infected animals and controls. For a broad representation of the spectrum of microglial morphology in each defined cluster, we chose representative cells of homeostatic and reactive microglia, using the sum of the distances of each cell in relation to all the others. Based on multivariate analysis, reactive microglia of infected animals showed more complex trees and thicker branches, covering a larger volume of tissue than in control animals. This approach offers a reliable representation of microglia dispersion in the Euclidean space, revealing the morphological kaleidoscope of surveillant and reactive microglia morphotypes. Because form precedes function in nature, our findings offer a starting point for research using integrative methods to understand microglia form and function.

## 1. Introduction

Viral encephalitis represents 20% to 50% of encephalitis cases in the United States. Among the known viral agents, the herpes simplex virus is responsible for 50% to 75% of these cases, with the remainder associated with varicella-zoster virus, enterovirus, or arbovirus [1]. In the past few decades, the geographical distribution of endemic arboviruses has expanded in Europe and caused an increasing number of outbreaks in humans [2]. Most arboviruses originally found in tropical regions, such as Africa and South America, or in some regions of Asia, are now widely dispersed and causing disease globally, including encephalitis. Several factors have contributed to the dispersion of arboviruses into new geographic areas, including global warming, increased urbanization, population growth in tropical regions, faster transport, and the rapid spread of arthropod vectors [3]. However, many of the mechanisms underlying the inflammatory response associated with virus encephalitis remain unknown [4,5], including those associated with important neurocognitive impairments related to arboviruses [6]. Microglia constitute approximately 7% of non-neuronal cells in different brain structures, as well as in the entire brain of many mammalian species [7], and their diversity of reactive phenotypes and contribution to maintaining central nervous system (CNS) homeostasis is remarkable [8]. For this reason, experimental studies focused on the adult host microglial response to arbovirus-associated encephalitis, such as the work described here, are essential to understanding its neuropathogenesis.

In addition to variable gene expression, the morphological features of microglia in the CNS vary significantly, including in ramifications and cell size, among species studied to date [9]. Although the homeostatic functions of microglia in the healthy CNS remain poorly defined and subject to considerable speculation [10], what is established is that microglial processes sweep the surrounding environment, monitoring the functional state of synapses, regulating neuronal activity in all stages of development and adult life, and participating in the remodeling and maturation of synaptic circuits through their contacts with pre- and post-synaptic membranes [11,12,13]; for a recent review, see [14].

In addition to the homeostatic microglial profile, there are multiple reactive microglial states, each making a specific physiological contribution, regulated by purinergic mechanisms. These mechanisms come into play because of discrete changes in the brain parenchyma in both physiological and pathological conditions [15]. The motility of microglia includes constant extension and retraction of the cellular processes to examine the brain, and directing processes to sites with tissue damage, which are controlled by P2Y12 activation and membrane potential [16]. The microglia also exhibit nanoscale surveillance with adaptation of filopodia and increased local concentration of cAMP. Fluctuating levels of intracellular cAMP control the polarity of microglial responses and alter the scale of surveillance work [17].

In mice exposed to lipopolysaccharide, microglial morphology is directly correlated with gene expression in the CNS, based on three-dimensional (3D) microscopic reconstruction and transcriptomic analysis in the same individuals [18]. In that work, the results in males and females demonstrated differential and significant correlation of the total area and volume of the immunolabeled microglial processes in CA3 with selective ionized calcium binding adapter molecule 1 (IBA1) and P2Y12 gene expression. Different classes of transcription factors interact to select and activate regulatory elements that control microglia responses to a variety of signals [19]. For example, microglia-specific disruption of *interferon regulatory factor 8* significantly reduces microglia ramification and surface area and alters the expression of several cell surface markers, shifting the microglia phenotype to a reactive profile [20].

Thus, the close association of microglial form and function is widely accepted. Because form precedes function [21,22], characterizing forms under different conditions is a good starting point for unraveling microglia morphotypes, with analysis of morphological shifts that follow on environmental changes.

Here, we used an arbovirus encephalitis model for this characterization. In a previous study, we induced Piry viral sublethal encephalitis in an adult albino Swiss mouse model and studied the resulting neuropathological damage and behavioral changes under controlled environmental conditions [23]. We found that Piry neurotropic neuroinvasion through the olfactory epithelium induces encephalitis characterized by inflammation along the limbic areas of the brain, including the olfactory pathways, septum, hippocampus, and amygdala, with rapid activation of microglia. We reconstructed microglia three dimensionally from the CA3 hippocampus field of control in uninfected and Piry virus–infected animals and demonstrated the utility of morphometry and hierarchical cluster analysis for distinguishing between the two conditions [24].

Three fundamental problems require attention before cells can be classified by their morphology in the CNS and other tissues: unambiguous identification with selective markers, definition of readily recognizable boundaries of the sampling area, and application of a random and systematic stereological sample approach to ensure that the reconstructed cell sample within these boundaries is free from bias [25,26,27,28]. Computerized microscopy systems allow for 3D reconstruction, recording the position of the soma and branches, tree origin, and process thickness; assigning an order to the traced points; identifying bifurcation and terminal points; differentiating natural terminations from cut ends; and merging tree segments from serial sections [29,30,31,32,33,34]. Reconstructed cells are then morphologically classified using multivariate statistical analysis of their morphometric features with hierarchical cluster, principal component, and discriminant analyses [35,36,37].

Here, we revisited the slide collections from our previous study of CA3 [23] and reconstructed microglia from the CA1/CA2 hippocampus field of the same animals, all maintained under standard laboratory conditions. We used a stereological approach for unbiased selection of microglia used in 3D microscopic reconstructions. Using multivariate statistical analysis of morphometric features and distance matrix analysis of both uninfected and infected groups, we searched for morphological changes and found many differential morphotypes between the two groups that may reflect underlying functional changes imposed by homeostatic disruption.

## 2. Experimental Procedures

In this work, 10 adult female Swiss albino mice at 12 weeks of age were used, originating from the colony of the Animal House of the Evandro Chagas-Pará Institute, and manipulated according to the “Principles of Laboratory Animal Care” of the National Health Institute (National Institutes of Health (NIH), Bethesda, MD, USA) and of the National Committee for Animal Experimentation (National Council of Animal Experimentation CONCEA, Ministry of Science, Technology and Innovation, Brazil). Use of these animals was approved in 2005 by the Research Ethics Committee with Experimental Animals (CEPAE-UFPA) registered under protocol number 1701/2005. The present investigation was carried out in compliance with relevant institutional biosafety and biosecurity international protocols adopted by the Evandro Chagas Institute.

The infected and control animals used in this work were limited to those from a previous investigation [23], all maintained in standard laboratory cages at 15 m above sea level [38]. All animals were sacrificed at 8 days after intranasal instillation of a suspension of infected or normal (uninfected) brain homogenate. To ensure clarity of the text, the methodology is briefly presented here, and a detailed description has been published elsewhere [23].

### 2.1. Animals and Infection

Animals were accommodated in standard laboratory cages (five animals per cage) and kept in a controlled temperature environment (22 °C), with 12-h cycles of light–dark, starting at 7:00 in the morning.

For intranasal inoculation, we used brain homogenates infected with the Piry virus or uninfected homogenates. The virus-infected brain homogenates were obtained from neonatal mice previously infected with the pathogen, provided by the Arbovirology and Hemorrhagic Fever Department following a previously described protocol [39]. Each brain (0.2 g) was macerated in 1.8 mL of 0.05 M phosphate-buffered saline (PBS; pH 7.2–7.4) containing penicillin (100 U/mL) and streptomycin (100 µg/mL). The mixture was centrifuged at 10,000× *g* for 15 min at 4 °C and the resulting suspension aliquoted into 0.5-mL tubes and kept at −70 °C until use.

Viral titration was performed by intra-cerebral inoculation of 0.2 mL of viral suspension in different PBS dilutions on a multiple scale of 10. The median lethal dose (LD50) was calculated by the method of Reed and Muench [40]. The initial LD50/20 µL was 8.0 Log10. The viral concentration chosen in this work was based on the need to obtain a non-lethal dose for the entire colony and one that could produce chronic non-fatal encephalitis. Finally, 20 µL of viral suspension in 0.1 M phosphate buffer solution, pH 7.2–7.4, containing penicillin (100 U/mL) and streptomycin (100 mg/mL) was administered intranasally.

The effective dose for each group of experiments varied based on evident clinical signs and deaths in <50%. The control group consisted of five animals that received 20 µL of uninfected neonate mouse brain homogenate.

### 2.2. Perfusion and Microtomy

The mice were anesthetized with Avertin (0.15 mL/5 g of body weight [41]), administered intraperitoneally and perfused intracardially with heparinized saline and 4% paraformaldehyde in 0.1 M phosphate buffer, pH 7.2–7.4. After craniotomy, the brains were kept in 2% paraformaldehyde for 24 h and then sectioned in the parasagittal plane at a thickness of 70 µm in a vibratome (MICROM, model HM 650 MV, Waldorf, Germany). Anatomical series of sections (1/4) obtained from each brain were divided into four samples equally representative of the whole brain for further processing by immunohistochemistry.

For 3D reconstructions, microglia from control and infected animals were immunolabeled with 2 µg/mL of an antibody to IBA-1, an adapter molecule that binds to calcium in macrophages/microglia and is specifically expressed in these cells. It is upregulated during microglia activation [42,43]. A detailed protocol for the immunostaining has been published elsewhere [24]. In brief, the anti-IBA-1 antibody (anti-Iba1, #019-19741; Wako Pure Chemical Industries Ltd., Osaka, Japan) was diluted in PBS (pH 7.2) and used to immunolabel an alternate series of sections to detect microglia and/or macrophages. For immunolabeling, free-floating sections were pre-treated with 0.2 M boric acid, pH 9, at 65–70 °C for 60 min to improve antigen retrieval, washed in 5% PBS, immersed for 20 min in 10% casein (Vector Laboratories, Burlingame, CA, USA), and then incubated with anti-Iba1 (2 µg/mL in PBS, diluted in 0.1 M PBS, pH 7.2–7.4), for 3 days at 4 °C with gentle and continuous agitation. Washed sections were then incubated overnight with secondary antibody (biotinylated goat anti-rabbit, 1:250 in PBS; Vector Laboratories, Burlingame, CA, USA). Endogenous peroxidases were inactivated by immersion of the sections in 3% H_2_O_2_/PBS, then washed in PBS and transferred to a solution of avidin-biotin-peroxidase complex (Vectastain ABC kit; Vector Laboratories, Burlingame, CA, USA) for 1 h. The sections were washed again before incubation in 0.1 M acetate buffer, pH 6.0, for 3 min, and developed in a solution of 0.6 mg/mL diaminobenzidine, 2.5 mg/mL ammonium nickel chloride, and 0.1 mg/mL glucose oxidase. After immunolabeling, all sections were counterstained by cresyl violet. Immunoreacted sections were mounted on gelatinized slides, dried at room temperature, and subsequently dehydrated in alcohols in increasing concentrations (70%, 80%, 90%, and 100%), diaphanized in xylol, and covered with a coverslip using Entellan^®^ (Merck KGaA, Darmstadt, Germany).

### 2.3. Morphometry Based on 3D Reconstruction

In sections stained by immunohistochemistry for IBA-1 (microglia) and counterstained by Nissl, the boundaries of CA1/CA2 were determined according to the cytoarchitecture of the parasagittal atlas of Nissl-stained sections [44], taking into account the size and degree of packaging of the cells as well as the lamination. For quantitative analysis, we included only cells with true branch ends within a single section. Cells with branches that seemed to have been cut during sectioning were not included. Thus, all cells with cut end branches extending to the top or bottom of a section under analysis were discarded.

For the 3D reconstruction of the microglia, we used an optical microscope (Eclipse 80i, Nikon, Nikon Instruments Inc., Konan, Minato-ku, Tokyo, Japan) with a motorized stage and analog-digital converters (MAC6000 System, Ludl Electronic Products, Hawthorne, NY, USA) to digitally store information related to spatial coordinates (X, Y, Z) of each point of the reconstruction. This system was coupled to a microprocessor that controlled the movements of the motorized stage with the aid of dedicated software (Neurolucida, MBF Bioscience, Williston, VT, USA). To avoid ambiguities in identifying objects of interest and to ensure greater precision in reconstructions, the 4.0× objective was replaced with a PLANFLUOR 100× objective (NA 1.3; DF = 0.2 µm; Nikon, Konan, Minato-ku, Tokyo, Japan) used for 3D reconstructions. As noted, only microglia showing complete branches were used for reconstruction. To ensure that all regions were equally likely to contribute to the sample, we adopted a random and systematic stereological approach to select microglia [26] from a series of sections containing CA1/CA2. We generated squared probes (50 × 50 µm) over a grid sample (200 × 200 µm) placed over the CA1/CA2 area of interest. From each squared probe, we selected single microglia for 3D reconstruction, removing the sample bias. For all experimental groups, we applied a correction for retraction induced by histological processing in the morphometric data, limited to the *z* axis, in a linear way, where 75% of shrinkage was assumed to occur, as previously suggested [45].

### 2.4. Statistical Analysis of Morphometry

Using alternate sections of the same anatomical series, we reconstructed 211 microglia immunostained for IBA1, 113 from infected, and 98 from control animals. Initially, we searched for shared morphometric features among the reconstructed microglia in our sample, within each experimental group. All quantitative morphometric variables with multimodality indexes (MMIs) >0.55 were selected for cluster analysis (Ward’s hierarchical clustering method), which included all animals in each group. Ward’s method relies on the minimum variance criterion to minimize within-cluster variance [46] and has been used to classify microglia [36] and other cells [35,47] by morphometric features. We estimated the MMI based on the asymmetry and kurtosis of our sample for each morphometric variable according to the equation: MMI = [M3^2^ + 1]/[M4 + 3 (n − 1)^2^/(n − 2) (n − 3)], where M3 is asymmetry, M4 is kurtosis, and n is the sample size. Kurtosis and asymmetry describe the form of data distribution and distinction among unimodal, bimodal, and multimodal curves [35,48].

The multimodal index of each variable was estimated based on the measurements of 20 morphometric parameters of the branches of microglia, as follows: total length of branches (μm), average length of branches (μm), average surface area of branches (μm^2^), volume of branches (μm^3^), total number of segments, segments per millimeter, tortuosity, fractal dimensions (k-dim), base diameter of primary branches (μm), total tree surface area (μm^2^), planar angle, number of trees, complexity, convex-hull volume (μm^3^), convex-hull surface area, convex-hull area, convex-hull perimeter, vertex va, vertex vb, and vertex vc (see Table 1 for details). Morphological characteristics of microglia with MMI > 0.55 indicated that their distribution was at least bimodal and could be multimodal [35].

MMI morphometric parameters with normalized scales were selected for the cluster analysis. We used Ward’s method with standardized variables and a tree diagram (dendrogram) to illustrate the classification generated by the cluster analysis. Ward’s Method uses the minimum variance criterion to minimize within-cluster variance [46] (Ward, 1963) and has been used to classify microglia [36] and other cells [35,47] using morphometric features.

To identify which variables contributed most to the formation of clusters, we subjected the data generated by the cluster hierarchical analysis to analysis of the canonical discriminating function, using the software Statistica 12.0 (StatSoft, Tulsa, OK, USA). The purpose of this procedure was to determine whether the groups differed in the mean of a variable and then to use that variable to predict group membership. To determine if there were significant differences between groups (over all variables), we used multivariate F tests for comparisons between matrices of total variances and covariances. In a step-forward discriminant function analysis, the program builds a step-by-step discrimination model. In this model, at each stage, all variables are reviewed and evaluated to determine which variable contributes most to discrimination between groups. We applied this procedure to determine the morphometric variables that provided the best separation between the classes of microglia suggested by the cluster analysis. In addition, we calculated the arithmetic mean and standard deviation for the variables chosen as the best predictors for the microglial groups. Parametric statistical analyzes with t-tests were applied to compare groups of microglia within each experimental group. All CA1/CA2 microglia of the hippocampal formation were measured several times, and dedicated software (Neurolucida Explorer, MBF Bioscience, Williston, VT, USA) was used to process the obtained data. We applied these procedures to our sample of microglia to look for potential morphological classes within each experimental group.

For morphometric analysis, as noted, we used 20 morphological measures for each of the cells reconstructed in three dimensions. These measures were adapted for the morphometry of the microglia using algorithms designed for morphological measures applied to 3D reconstructions of neurons. The definitions for each variable are shown in Table 1. We used STATISTICA software, version 12. StatSoft, Inc. (2014) to obtain the Euclidean distance matrices (distance between microglia pairs) and the sum of distances of each cell in relation to all the others.

## 3. Results

Overall, the results showed that the reactive microglia of infected animals exhibited more complex trees and a greater number of thicker branches that covered a larger volume of tissue than the microglia of control animals.

### Microglial Metamorphosis: From Surveillance to Reactivity to Virus Encephalitis

Immunostaining for Piry virus antigens in the mouse brain after 8 days of nasal instillation of an infectious suspension is illustrated in Figure 1. Virus antigens in cell targets were found in the cytoplasm of infected cells along the olfactory pathways. For further details and pictorial documentation, see [36].

Figure 2 shows the cluster analysis dendrogram (Figure 2A); the 3D reconstructions of microglia corresponding to each of the three identified clusters, two for infected animals (clusters 1 and 2) and a third for controls (cluster 3) (Figure 2B–D); and the discriminant function analysis results (Figure 2E). The microglial representative cells of CA1/CA2 in Figure 2B illustrate three morphotypes per cluster. For the choice of an “average cell” that illustrates each morphotype, we used the distance matrix to obtain the sum of the distances of each cell relative to all others. The matrices were constructed with the combination of all cells of a given group taken pairwise, followed by the weighted calculation of a scalar Euclidean distance between cells using all morphometric variables.

To show the morphological kaleidoscope of each cluster, we identified the cells with the smallest (top), median (middle), and largest (bottom) sum of the distances. With this algorithm, a greater morphological variation of microglia in the CA1/CA2 became apparent in infected animals as compared with controls. Thus, from the analysis depicted in Figure 2, nine contrasting morphotypes (three per cluster) were detected, revealing diverse morphological microglial phenotypes. In addition, we found that when grouping similar cells based on their morphometry, the multivariate quantitative analysis could obscure some of this morphological diversity if the choice of a group’s most representative cell was based only on the least sum of distance between cells. Note that the microglia representative of the infected animals (cluster 1, blue/#; cluster 2, green/+) showed a greater number of thicker branches covering more tissue and connected to larger cell bodies than microglia of the control group (cluster 3, orange/*).

Figure 2E depicts the results of the canonical discriminant function analysis performed with the groups indicated by the cluster analysis, based on the morphometric variables that most contributed to the cluster formation. Note that the cluster 1, blue/#; cluster 2, green/+ and cluster 3, orange/*, corresponding to each experimental group preserve the colors originally used in the hierarchical cluster analysis. Each color/shape occupies distinct regions of the Euclidean space, with minimal overlap between clusters 1 and 2 and clusters 1 and 3. No overlap was observed between clusters 2 and 3. Note that microglia of the control group (cluster 3) were less dispersed, suggesting greater morphological similarity among them. The separation between the centroids is greater between the control group (cluster 3) and cluster 2 of the infected animals, with the latter containing microglia with the greatest tree expansion. Function 1 of the discriminant analysis corresponded to 91.8% of the variance and included the surface of the convex hull, the total volume of the branches, and the morphological complexity as morphometric variables that contributed most to the variance (see Appendix A).

Figure 3 shows the mean values with the corresponding standard deviations and standard errors for the surface of the convex hull surface and for the total branch volume of each cluster. The convex hull data measures the size of the tissue volume covered by the microglial trees. It interprets a branching structure as a solid 3D object controlling a certain amount of physical space. The amount of physical space is defined in terms of the volume of tissue covered by the tree of a single microglia. The convex hull surface illustrated in Figure 3 represents the sum of the areas of the faces of the corresponding irregular convex polyhedron defined by microglial branches.

The analysis depicted in Figure 3 makes clear that Piry virus encephalitis induced significant expansion of microglial trees and the total volume of microglial branches.

Figure 4 shows mean values with the corresponding standard deviations (whiskers) and errors (box edges) of the morphological complexity and the branch volume of each cluster. In general, the lowest mean values for these morphometric variables corresponded to the microglia of the control animals (orange/*). The highest (green/+) and intermediate values (blue/#) were from microglia of infected animals.

The definitions of morphological complexity (Figure 3) are based on the following equation: Complexity = [Sum of the terminal orders + Number of terminals] × [Total branch length/Number of primary branches]. It can be roughly inferred that this morphometric indicator analyzes in an integrated manner a series of parameters of the microglial tree that, together with the volume of the convex hull, suggested significant dissimilarity of microglial features between infected and control animals.

Figure 5 illustrates in a 3D Cartesian representation the values of the three variables that most contributed to the formation of clusters from control (orange/*) and infected (green/+ and blue/#) animals. On the *y* axis are the values for the convex hull surface area. The total volume of the branches is on the *x* axis, and the *z* axis contains the values for morphological complexity.

Table 2 lists the values of the descriptive statistics for the morphometric variables, indicated by the analysis of the discriminant function, accounting for 91.8% of the variance.

Taken together, the numbers show that the microglia of the control animals had smaller trees that covered less tissue volume, with thinner branches and less morphological complexity.

In contrast, the reactive microglia had a more complex tree and thicker branches, which covered a greater volume of tissue. The analysis of variance shown in Table 3 confirms that these morphological differences were highly significant.

Table 4 shows the results after correction for multiple comparisons (Tukey’s HSD and Bonferroni) for the same morphometric variables. With the exception of tortuosity, all morphometric variables were significantly different between cluster 3 and clusters 1 and 2, distinguishing controls from infected animals and two groups of infected animals.

## 4. Discussion

In previous work with hippocampal CA3, we investigated the influence of age and environment on microglial morphology in correlation with disease progression [24]. Here, using the same Piry arbovirus model of experimental encephalitis in adult female Swiss albino mice, we expanded to hippocampal CA1/CA2, using the anatomical series of brain sections from that work. With a stereological sampling approach, we reconstructed microglia in three dimensions at day 8 post-infection. With hierarchical cluster analysis followed by linear discriminant function analysis, we classified the microglia of CA1/CA2 from control and infected mice and compared their morphology and morphometry. The results highlighted three large morphological clusters, two for microglia from infected animals and one representing control animal microglia. To identify the cells that best represented the morphological kaleidoscope of each cluster, we estimated the smallest (top), median (middle), and largest (bottom) sum of the distances. The matrices were constructed using all morphometric variables with the combination of all cells in a given group, followed by the weighted calculation of a scalar Euclidean distance between the cells. This approach revealed considerable morphological variation in the CA1/CA2 microglia of infected and control animals. Indeed, we found nine contrasting morphotypes (three for each identified cluster), revealing morphological aspects of microglia that otherwise would have been obscured by standard multivariate analysis.

### 4.1. Piry Virus Neuroinvasion and Microglial Response

Although most viral infections are concentrated in peripheral tissues (non-neurotropic viruses), many viral species reach the CNS, where they can alter homeostasis and induce neurological dysfunction associated with life-threatening inflammatory diseases [5,49,50,51].

The Piry arbovirus was previously chosen as a model for experimental studies because of the low level of biosafety required to work with it and its affiliation with the South American group of RNA viruses, occurrence in Brazil [52,53], and ability to produce experimental encephalitis in neonatal and adult mice [23,54,55,56]. In humans, it is associated with mild febrile disease [57,58].

Neurotropic viruses can use the olfactory system to invade the mammalian CNS, which is the pathway used by the herpes simplex virus 1 [59], mouse hepatitis virus [60], pseudorabies virus [61], Venezuelan equine encephalitis virus [62], and the CVS variety of the rabies virus [63]. After infecting the cerebral parenchyma, following neuroinvasion through the olfactory nerves, Piry virus antigens are found to be concentrated in the limbic areas, including olfactory bulb, septum, and hippocampus [23].

If we assume that form and function are intertwined and that morphology precedes function both in developing [64] and mature brain circuits [65], morphometric analysis of microglia either from homeostatic or altered conditions can open up several lines of questioning [37,66,67]. In the present work, with the help of cluster analysis, we demonstrated that in control animals kept in standard laboratory cages, the microglia of infected animals responded with significantly diversified morphology in the face of encephalitis induced by Piry arbovirus. This response included increased complexity, increased volume of tissue covered by branches, and an increased branch number and thickness. For complexity, as defined previously [68], an increasing value has been associated with hyper-ramified and longer processes even in the absence of inflammation or neurodegeneration in the prefrontal cortex of rats subjected to chronic stress [69,70]. Chronic stress significantly increases microglial branching complexity and enhances ramification, maintaining the area occupied by the cell. Of note, mice subjected to chronic stress express tumor necrosis factor (TNF)α, albeit at relatively low levels, and exhibit activation and pro-ramifying microglia effects in the prefrontal cortex [71]. Likewise, a significant increase in TNFα immunostaining has been demonstrated in association with an intense microglial response with branch thickening and increased morphological complexity in a model of single dengue viral infection [72] and with an exacerbated inflammatory response induced by heterologous antibody-enhancement dengue disease [73,74].

Some groups using different experimental models have previously documented increased β-integrin expression in association with the microglial response in altered CNS, with increased microglial morphological diversity, branch hypertrophy, and an increased number and thickness of branches [75,76]. In viral infections, a comparison of transcriptomic changes in astrocytes and microglia after innate immune stimulation by viral infections allows identification of glial activation markers [77]. Morphological analysis of the microglia shows that signaling by interferon (IFN) through the type I neuron receptor has a stronger impact on activating cells of myeloid origin (microglia) than on astrocytes. In addition to IFN-β induction, CNS infection by the olfactory pathway results in the recruitment in the CNS of myeloid cells, such as microglia [78], and peripheral immune cells, such as dendritic cells [79], T cells, and monocytes [23,80]. Cross-talk among neurons, astrocytes, and microglia is critical for complete microglia activation and protection against lethal encephalitis [81].

### 4.2. Reactive Microglial Morphology

With multiple processes that expand and retract continuously, guided by the sensome [82], microglia represent the most important cellular component of the innate immune response in the CNS [10,83,84]. Morphological studies in fixed tissue, such as the current work, in general allow for an instantaneous view on the microglial morphology of a given region associated with different experimental conditions and time windows. In the present report, we classified 3D-reconstructed microglia from fixed tissue of infected and control animals, using hierarchical cluster analysis and linear discriminant function analysis of multimodal morphometric features. We found three main clusters, two in infected animals and one in control animals.

Microglia and macrophages have been described classically as capable of generating two main types of response. The first is M1, which has a pro-inflammatory profile and can be mediated by TNF-alpha and IFN-gamma. The second is M2, an alternative anti-inflammatory pathway that acts in the resolution of inflammation and tissue repair and can be mediated by interleukin-4 [85]. However, more recently, much discussion has centered on the existence of several other reactive microglia phenotypes in addition to those recognized as M1 and M2, calling this binary classification into question [86]. Although microglia under homeostatic conditions have delicate, branching extensions oriented radially from a small elliptical soma [84,86], they exhibit a variety of forms that can undergo forward and reverse morphofunctional changes, supporting distinct neuroimmune functions in both homeostatic and non-homeostatic conditions [66,86,87,88,89]. Indeed, in almost all brain diseases, pathological progression is influenced by microglial cells [84] that react with the development of multiple morphofunctional phenotypes [90,91] as a function of age, environment (e.g., lifestyle), and the nature of the insult [92]. The multivariate statistical analysis of morphometric features of 3D-reconstructed microglia from homeostatic (control) and non-homeostatic (Piry virus–infected) mice in the current work revealed a kaleidoscope of microglial morphologies.

## 5. Conclusions

In this work, comparison of morphometric measurements of reactive microglia from Piry virus–infected animals and uninfected controls showed more complex trees and thicker branches covering a greater tissue volume in the infected animals. Thus, the relative energy cost per monitored unit of tissue volume through vigilant and reactive microglia may impose a metabolic trade-off between the neural (surveillance) and immunological (reactive) functions of these cells. Using an unbiased stereological approach to select microglia for hierarchical cluster and discriminant analyses, we identified a kaleidoscope of microglial morphologies. Given that in nature, form precedes function, our findings are a good starting point for research into integrative approaches for understanding microglial form and function.

## Figures and Tables

**Figure 1 brainsci-11-01009-f001:**
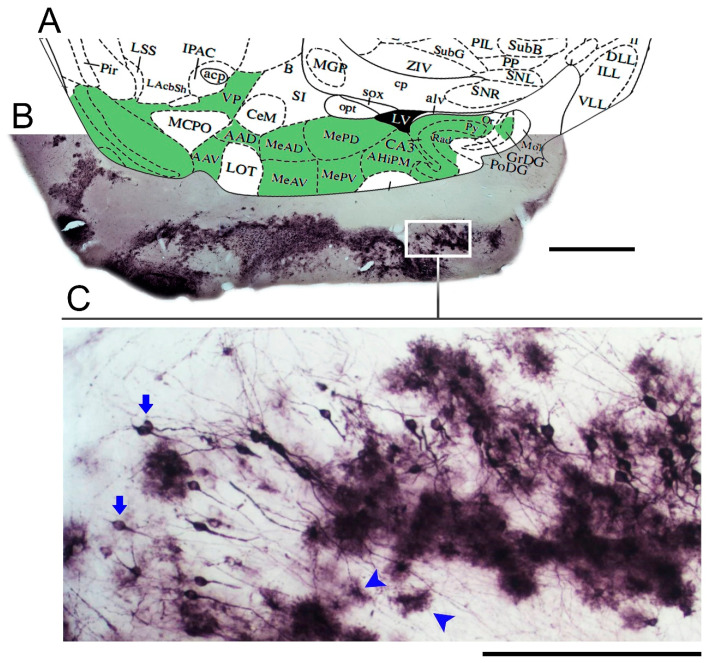
Low- and medium-power photomicrographs of a parasagittal section from the brain of a mouse infected with Piry virus, at day 8 post-nasal instillation. Diagram at the corresponding level (**A**) of a mouse brain stereotaxic atlas [44] is used to identify the stained neuroanatomical areas (green areas in the diagram) with virus antigens (**B**). Immunostained neurons and glial cells are indicated by arrows and arrow heads, respectively (**C**). VP = ventral pallidum; AAD = anterior amygdaloid area; AAV = anterior amygdaloid area—ventral part; MeAD = medial amygdaloid nucleus, anterior dorsal; MeAV = medial amygdaloid nucleus, anteroventral part; MePD = medial amygdaloid nucleus, posterodorsal part; MePV = medial amygdaloid nucleus, posteroventral part; CA3 = field CA3 of hippocampus; amygdalohippocampal area; Or = oriens layer of hippocampus; Py = pyramidal cell layer of hippocampus; Rad = Stratum radiatum of the hippocampus; Mol = molecular layer of dentate gyrus; GrDG = granular layer of dentate gyrus; PoDG = Polymorph layer of the dentate gyrus.

**Figure 2 brainsci-11-01009-f002:**
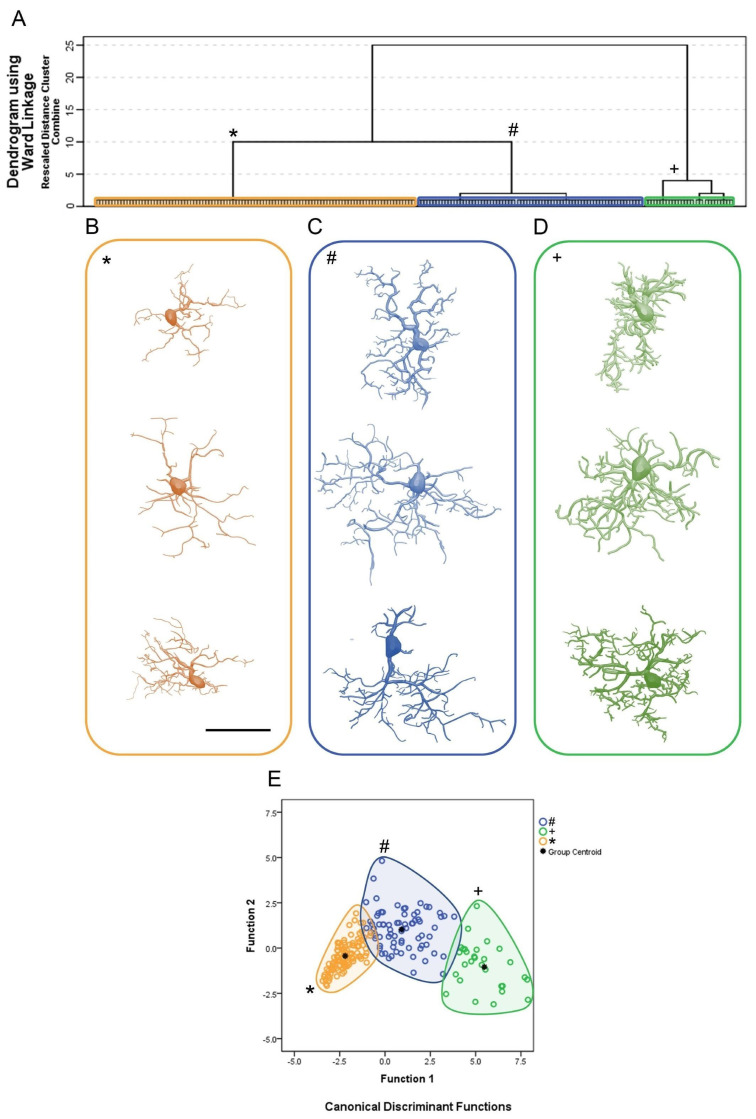
(**A**): Hierarchical cluster analysis for morphological classification of microglia from CA1/CA2 of control and Piry virus–infected animals at 8 days post-infection. Note the three main clusters, two of them representing microglia of infected animals (cluster 1, blue/#; cluster 2, green/+) and another representing microglia of the control group (cluster 3, orange/*). The morphometric variables used in the analysis were those with a MMI above 0.55 with at least bimodal distribution; (**B**–**D**): Representative cells of each cluster based on the greatest, median, and shortest distances between pairs, after multiple comparisons. The cells that best represent each cluster were based on the values of the shortest (upper row), median (intermediate), and the greatest (bottom row) sum of distances. (**E**): Canonical discriminating function analysis illustrating the distribution of microglia in the Euclidean space as a function of the clusters identified by the hierarchical cluster analysis. Note that the distribution of microglia in the Euclidean space of control animals (orange/*) is much more compact than the distribution of microglia of infected animals (blue/# and green/+) suggesting greater homogeneity of morphology among controls. The variables that most contributed to the formation of clusters were the surface of the convex hull, the total volume of the branches, and the complexity. For detailed information on the statistical analysis, see Appendix A.

**Figure 3 brainsci-11-01009-f003:**
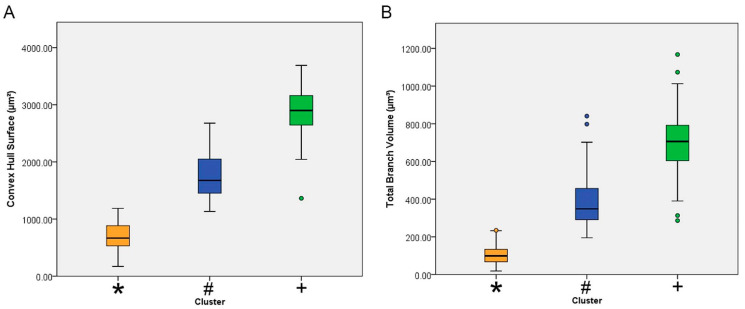
Mean values and corresponding standard errors (box edges) and standard deviations (whiskers) for the surface of the convex hull (**A**) and total branch volume (**B**) for microglia of Figure 1. Infected animals correspond to clusters 1(blue/#) and 2 (green/+) and control animals to cluster 3 (orange/*). Note that when compared to the microglia of the control animals, the mean value of the convex hull surface area of the microglia of the infected animals was 3 to 6-fold greater.

**Figure 4 brainsci-11-01009-f004:**
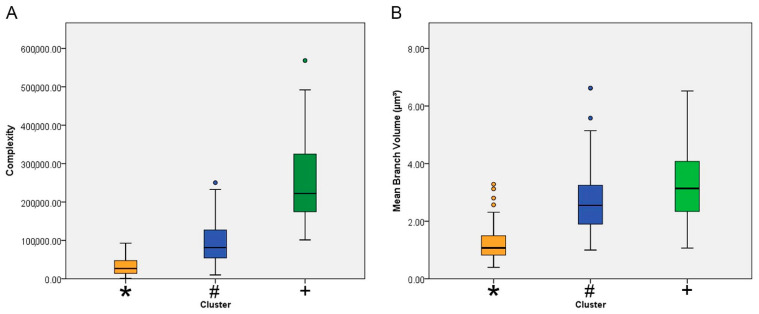
Mean values and corresponding errors (box edges) and standard deviations (whiskers) for the complexity (**A**) and the average volume of the microglia branches (**B**) of clusters 1 (blue/#) and 2 (green/+), both representing microglia from infected animals, and for control animal microglia (cluster 3, orange/*). Note that control animal microglia were less complex and exhibited thinner branches than the microglia of infected animals.

**Figure 5 brainsci-11-01009-f005:**
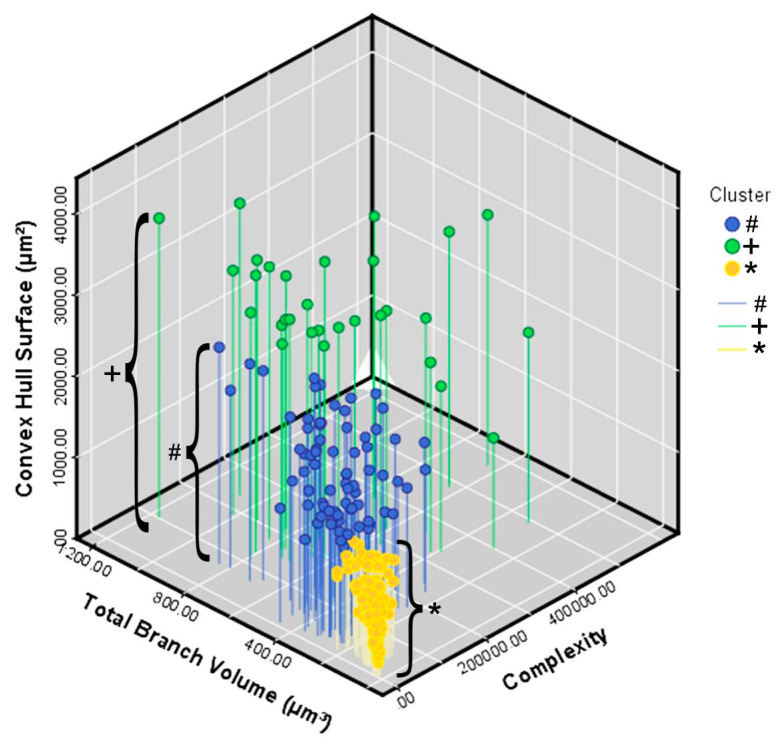
Three-dimensional graphic representation of the mean values of morphometric variables that most contributed to the cluster formation. The values of the surface area of the convex hull are indicated on the *y* axis, the values of the total volume of the branches are indicated on the *x* axis and the morphological complexity on the *z* axis. Note that the lower mean values on the three morphometric variables correspond to the orange filled circles indicated by brackets which indicate control animals and the greatest ones correspond to group 2 of the infected animals (+bracket) indicated by filled green circles on top of green lines.

**Table 1 brainsci-11-01009-t001:** Morphological measurements of microscopic 3D reconstructions.

Branched Structure Analysis	
Segment	Any portion of microglia branched structure with endings that are either nodes or terminations with no intermediate nodes
Segments/mm	Number of segments/total length of the segments, expressed in millimeters
No of trees	Number of trees in the microglia
Total No of segments	Total number of segments in the tree
Single branch length	Total length of the line segments used to trace the branch of interest
Mean branch length (µm)	Mean = [Total length]/[Number of branches]
Total branch length (µm)	Total length for all branches in the tree
Tortuosity	Tortuosity = [Actual length of the segment] /[Distance between the endpoints of the segment]; smallest value is 1, which represents a straight segment; tortuosity allows segments of different lengths to be compared in terms of the complexity of the paths they take
Mean branch surface area (µm^2^)	Computed by modeling each branch as a frustum (truncated right circular cone) divided by the number of branches
Total tree surface area (µm^2^)	Two-dimensional (2D) surface area of a microglia arbor computed based on the area defined by the endpoints of all trees
Branch volume (µm^3^)	Computed by modeling each piece of each branch as a frustum
Total branch volume	Total volume for all branches in the tree
Base diameter of primary branch (µm)	Diameter at the start of the first segment
Planar angle	Computed based on the endpoints of the segments; references the change in direction of a segment relative to the previous segment
Fractal dimension k-dim	The “k-dim” of the fractal analysis, describing how the structure of interest fills space; significant statistical differences in k-dim suggest morphological dissimilarities
Convex hull: perimeter (µm), area (µm^2^), 2D surface area (µm^2^), 3D or volume (µm^3^)	Convex hull measures the size of the branching field by interpreting a branched structure as a solid object controlling a given amount of physical space; the amount of physical space is defined in terms of convex-hull volume, surface area, area, and/or perimeter.
Vertex analysis	Describes the overall structure of a branched object based on topological and metrical properties. Root (or origin) point: For neurons, microglia or astrocytes, the origin is the point at which the structure is attached to the soma. Main types of vertices: V_d_ (bifurcation) or V_t_ (trifurcation), nodal (or branching) points. V_p_: Terminal (or pendant) vertices. V_a_: primary vertices connecting 2 pendant vertices; V_b_: secondary vertices connecting 1 pendant vertex (V_p_) to 1 bifurcation (V_d_) or 1 trifurcation (V_t_); V_c_: tertiary vertices connecting either 2 bifurcations (V_d_), 2 trifurcations (V_t_), or 1 bifurcation (V_d_) and 1 trifurcation (V_t_). In the present report, we measured the number of vertices Va, Vb, and Vc.
Complexity	Complexity = [Sum of the terminal orders + Number of terminals] × [Total branch length/Number of primary branches]

**Table 2 brainsci-11-01009-t002:** Descriptive statistical results for morphometric variables that contributed most to the formation of clusters. The data for the clusters are shown as 1, 2, and 3 on each line. Clusters 1 and 2 correspond to the microglia of the infected animals, and cluster 3 represents microglia of control animals. Note that the lowest mean values for each variable were in cluster 3 (control animals) and the highest values in cluster 2 (one group of infected animals).

Descriptive Results									
		N	Mean	Std. Deviation	Std. Error	95% Confidence Interval for Mean		Minimum	Maximum
						Lower Bound	Upper Bound		
Tortuosity	1	75	1.33	0.12	0.01	1.30	1.36	1.09	1.56
	2	30	1.35	0.09	0.02	1.31	1.38	1.18	1.59
	3	106	1.20	0.09	0.01	1.18	1.21	1.08	1.66
	Total	211	1.27	0.12	0.01	1.25	1.28	1.08	1.66
Total branch volume (µm^3^)	1	75	381.99	131.21	15.15	351.80	412.17	194.82	840.86
	2	30	696.56	202.32	36.94	621.01	772.11	286.29	1167.60
	3	106	104.39	52.92	5.14	94.20	114.58	18.69	234.52
	Total	211	287.26	239.03	16.46	254.82	319.70	18.69	1167.60
Mean branch volume (µm^3^)	1	75	2.79	1.13	0.13	2.53	3.05	1.00	6.62
	2	30	3.34	1.36	0.25	2.83	3.85	1.07	6.52
	3	106	1.22	0.55	0.05	1.12	1.33	0.40	3.28
	Total	211	2.08	1.28	0.09	1.91	2.25	0.40	6.62
Convex hull surface (µm^2^)	1	75	1768.35	392.52	45.32	1678.04	1858.66	1133.26	2677.56
	2	30	2882.78	473.73	86.49	2705.89	3059.67	1362.55	3689.12
	3	106	689.23	247.31	24.02	641.60	736.86	170.98	1186.62
	Total	211	1384.68	856.26	58.95	1268.48	1500.89	170.98	3689.12
Complexity	1	75	945,72.82	53,188.84	6141.72	82,335.18	106,810.46	10,038.40	250,383.00
	2	30	261,064.33	117,350.51	21,425.17	217,244.93	304,883.73	101,354.00	568,527.00
	3	106	31,720.23	21,318.02	2070.59	27,614.64	35,825.83	1451.06	92,769.60
	Total	211	86,669.37	94,958.17	6537.19	73,782.44	99,556.29	1451.06	568,527.00

**Table 3 brainsci-11-01009-t003:** Results of the one-way analysis of variance (ANOVA) indicating significance (Sig.) and respective F values from group comparisons.

One-Way ANOVA						
		Sum of Squares	df	Mean Square	F	Sig.
Tortuosity	Between groups	1.01	2.00	0.50	51.64	0.00
	Within groups	2.03	208.00	0.01		
	Total	3.03	210.00			
Total branch volume (µm^3^)	Between groups	9,243,466.58	2.00	4,621,733.29	348.92	0.00
	Within groups	2,755,100.53	208.00	13,245.68		
	Total	11,998,567.12	210.00			
Mean branch volume (µm^3^)	Between groups	163.21	2.00	81.61	94.20	0.00
	Within groups	180.20	208.00	0.87		
	Total	343.41	210.00			
Convex hull surface (µm^2^)	Between groups	129,635,754.77	2.00	64,817,877.38	554.10	0.00
	Within groups	24,331,445.08	208.00	116,978.10		
	Total	153,967,199.85	210.00			
Complexity	Between groups	1,237,150,125,189.31	2.00	618,575,062,594.66	196.00	0.00
	Within groups	656,431,092,622.02	208.00	3,155,918,714.53		
	Total	1,893,581,217,811.34	210.00			

**Table 4 brainsci-11-01009-t004:** Values for morphometric variables of CA1/CA2 microglia, adjusted for multiple comparisons. Except for tortuosity, all morphometric variables differed significantly between cluster 3 (controls) and clusters 1 and 2 (infected animals).

Multiple Comparisons								
Dependent Variable	Between-Groups Comparisons			Mean Difference (I-J)	Std. Error	Sig.	95% Confidence Interval	
							Lower Bound	Upper Bound
Tortuosity	Tukey HSD	1 vs.	2	−0.017	0.021	0.710	−0.067	0.034
			3	0.13291 *	0.015	0.000	0.098	0.168
		2	1	0.017	0.021	0.710	−0.034	0.067
			3	0.14976 *	0.020	0.000	0.102	0.198
		3	1	−0.13291 *	0.015	0.000	−0.168	−0.098
			2	−0.14976 *	0.020	0.000	−0.198	−0.102
Total branch volume (µm^3^)	Tukey HSD	1 vs.	2	−314.57115 *	24.862	0.000	−373.261	−255.881
			3	277.59531 *	17.366	0.000	236.602	318.589
		2	1	314.57115 *	24.862	0.000	255.881	373.261
			3	592.16647 *	23.801	0.000	535.982	648.351
		3	1	−277.59531 *	17.366	0.000	−318.589	−236.602
			2	−592.16647 *	23.801	0.000	−648.351	−535.982
Mean branch volume (µm^3^)	Tukey HSD	1 vs.	2	−0.55013 *	0.201	0.018	−1.025	−0.075
			3	1.56653 *	0.140	0.000	1.235	1.898
		2	1	0.55013 *	0.201	0.018	0.075	1.025
			3	2.11666 *	0.192	0.000	1.662	2.571
		3	1	−1.56653 *	0.140	0.000	−1.898	−1.235
			2	−2.11666 *	0.192	0.000	−2.571	−1.662
Convex hull surface (µm^2^)	Tukey HSD	1 vs.	2	−1114.43247 *	73.885	0.000	−1288.846	−940.019
			3	1079.11491 *	51.607	0.000	957.291	1200.939
		2	1	1114.43247 *	73.885	0.000	940.019	1288.846
			3	2193.54738 *	70.731	0.000	2026.579	2360.515
		3	1	−1079.11491 *	51.607	0.000	−1200.939	−957.291
			2	−2193.54738 *	70.731	0.000	−2360.515	−2026.579
Complexity	Tukey HSD	1 vs.	2	−166,491.51467 *	12,135.741	0.000	−195,139.323	−137,843.707
			3	62,852.58386 *	8476.540	0.000	42,842.739	82,862.429
		2	1	166,491.51467 *	12,135.741	0.000	137,843.707	195,139.323
			3	229,344.09852 *	11,617.664	0.000	201,919.271	256,768.926
		3	1	−62,852.58386 *	8476.540	0.000	−82,862.429	−42,842.739
			2	−229,344.09852 *	11,617.664	0.000	−256,768.926	−201,919.271

* mean difference significant at the 0.05 level.

## Data Availability

The dataset(s) supporting the conclusions of this article are included within the article.

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
