# Peer review of "Microglial Metamorphosis in Three Dimensions in Virus Limbic Encephalitis: An Unbiased Pictorial Representation Based on a Stereological Sampling Approach of Surveillant and Reactive Microglia"

_brainsci, 2021, doi:10.3390/brainsci11081009_

Round 1

Reviewer 1 Report

The authors in the current study present computational analyses of microglia morphology in Piry virus or uninfected mouse brains from a previous study (de Sousa et al., 2011). They report that they can classify Piry infected animals from uninfected animals based on a few metrics on microglial morphology. The study is novel in that they approach clustering and identification of infection in a data driven way. Although this study is performed well some suggestions and comments are provided below.  

  1. Introduction - The main thrust of the paper is computational in nature. The introduction doesn't elaborate or give enough background on computational methods used to elucidate functions in the nervous system i.e. neurons, microglia, astrocytes. The authors need to provide more background on computational methods used to understand how structure precedes or even informs function.

2. Experimental Procedures

2.1 Animals and infection

1. It isn't clear how old the mice are given that in one paragraph it was described that "female Swiss albino mice with 2 months of age were used," and two paragraphs later "Ten adult female albino Swiss mice (Mus musculus) at 12 weeks of age were obtained". These two ranges are not similar. Could the authors clarify the age of the mice used in the current study.

2. Could the authors provide additional information as to the environmental enrichment status of the 10 animals used. The reference provided by the authors distinctly studied the effects of environmental enrichment on microglial activation in different areas of the brain including the CA3 of the hippocampus.  It isn't clear whether these 10 animals were from impoverished or enriched environments. Please indicate.

3. Could the authors provide the altitude of the animal facility that these animals were housed as there are effects of altitude on brain function (Kanekar et al., 2018).

3. Results

4. In Figure 1, the authors may consider using an alternative color besides green to show in Panel D. This is to accommodate color blindness individuals. The color scheme would then follow into the remaining Figures 2-4.

5. Figure 1, Panel E, the authors may consider using different shapes besides every group being circles. For black and white prints, this figure would lose information.

Discussion

The authors bring up the M1 and M2 binary distinctions in microglia and present readers with a spectrum of morphological differences to classify microglia. The authors should include paragraph on how microglia are currently classified versus how the authors would like to have readers approach microglia. 

It isn't clear why the authors discuss the role of ATP and purinergic receptors in microglia motility if they did not measure either in the study. The authors should either elaborate on how ATP and P2Y12 receptors are related to morphology and how this relates to function, or elaborate further on how structure precedes function. 

5.Conclusions

It isn't clear what the metabolic implications are that come from this algorithm that the authors have described. Could the authors elaborate further on the metabolic implications. 

Could the authors clarify "we identified the representative cells of each cluster using the largest, the median and
the smallest sum of the scalar distances using all the other cells for comparisons." what the all other cells are?

Author Response

Reviewer`s 1 comments and replies.

Comments and Suggestions for Authors

The authors in the current study present computational analyses of microglia morphology in Piry virus or uninfected mouse brains from a previous study (de Sousa et al., 2011). They report that they can classify Piry infected animals from uninfected animals based on a few metrics on microglial morphology. The study is novel in that they approach clustering and identification of infection in a data driven way. Although this study is performed well some suggestions and comments are provided below. 

Introduction - The main thrust of the paper is computational in nature. The introduction doesn't elaborate or give enough background on computational methods used to elucidate functions in the nervous system i.e. neurons, microglia, astrocytes. The authors need to provide more background on computational methods used to understand how structure precedes or even informs function.

Reply:  As suggested we now provided more background on computational methods used to classify cells as follows: “…Three fundamental problems require attention before cells can be classified by their morphology in the CNS and other tissues: unambiguous identification with selective markers, definition of readily recognizable boundaries of the sampling area, and application of a random and systematic stereological sample approach to ensure that the reconstructed cell sample within these boundaries is free from bias [25-28]. Computerized microscopy systems allow for 3D reconstruction, recording the position of the soma and branches, tree origin, and process thickness; assigning an order to the traced points; identifying bifurcation and terminal points; differentiating natural terminations from cut ends; and merging tree segments from serial sections [29-34]. Reconstructed cells are then morphologically classified using multivariate statistical analysis of their morphometric features with hierarchical cluster, principal component, and discriminant analyses [35-37]. …”

  1. Experimental Procedures

2.1 Animals and infection

  1. It isn't clear how old the mice are given that in one paragraph it was described that "female Swiss albino mice with 2 months of age were used," and two paragraphs later "Ten adult female albino Swiss mice (Mus musculus) at 12 weeks of age were obtained". These two ranges are not similar. Could the authors clarify the age of the mice used in the current study.

Reply: As suggested we now clarify the age of the mice in the current study.

  1. Could the authors provide additional information as to the environmental enrichment status of the 10 animals used. The reference provided by the authors distinctly studied the effects of environmental enrichment on microglial activation in different areas of the brain including the CA3 of the hippocampus. It isn't clear whether these 10 animals were from impoverished or enriched environments. Please indicate.

Reply: Although the reference provided (de Sousa et al., 2011) distinctly studied the effects of environment on sublethal encephalitis outcomes, in this report we have used mice from standard laboratory cages only. This information is now available in Experimental Procedures as follows: “…The infected and control animals used in this work were limited to those from a previous investigation [23], all maintained in standard laboratory cages at 15 m above sea level [38]…”

  1. Could the authors provide the altitude of the animal facility that these animals were housed as there are effects of altitude on brain function (Kanekar et al., 2018).

Reply: As suggested we now provided the altitude of the animal facility as follows: “…The infected and control animals used in this work were limited to those from a previous investigation [23], all maintained in standard laboratory cages at 15 m above sea level [38]…”

  1. Results

  1. In Figure 1, the authors may consider using an alternative color besides green to show in Panel D. This is to accommodate color blindness individuals. The color scheme would then follow into the remaining Figures 2-4.

Reply: Although we maintained original colors of Figure 1, we now distinguished experimental groups with different symbols and indicate with capital letters and numbers, cluster dendrogram and cells of different groups to accommodate color blindness individuals.

  1. Figure 1, Panel E, the authors may consider using different shapes besides every group being circles. For black and white prints, this figure would lose information.

Reply: Although we maintained original colors of Figure 1, we now distinguished experimental groups with different symbols and indicate with capital letters and numbers, cluster dendrogram and cells of different groups to accommodate color blindness individuals.

Discussion

The authors bring up the M1 and M2 binary distinctions in microglia and present readers with a spectrum of morphological differences to classify microglia. The authors should include paragraph on how microglia are currently classified versus how the authors would like to have readers approach microglia.

Reply: As suggested we now included a paragraph on how microglia are currently classified versus how we would like to have readers approach microglia as follows: “… Although microglia under homeostatic conditions have delicate, branching extensions oriented radially from a small elliptical soma [85,87], they exhibit a variety of forms that can undergo forward and reverse morphofunctional changes, supporting distinct neuroimmune functions in both homeostatic and non-homeostatic conditions [67,87-90]. Indeed, in almost all brain diseases, pathological progression is influenced by microglial cells [85] that react with the development of multiple morphofunctional phenotypes [91,92] as a function of age, environment (e.g., lifestyle), and the nature of the insult [93]. The multivariate statistical analysis of morphometric features of 3D-reconstructed microglia from homeostatic (control) and non-homeostatic (Piry virus–infected) mice in the current work revealed a kaleidoscope of microglial morphologies.

It isn't clear why the authors discuss the role of ATP and purinergic receptors in microglia motility if they did not measure either in the study. The authors should either elaborate on how ATP and P2Y12 receptors are related to morphology and how this relates to function, or elaborate further on how structure precedes function.

Reply: As suggested we deleted the paragraph related the role of ATP and purinergic receptors.

5.Conclusions

It isn't clear what the metabolic implications are that come from this algorithm that the authors have described. Could the authors elaborate further on the metabolic implications.

Could the authors clarify "we identified the representative cells of each cluster using the largest, the median and

the smallest sum of the scalar distances using all the other cells for comparisons." what the all other cells are?

Reply: As suggested we changed the conclusion paragraph as follows: “…In this work, comparison of morphometric measurements of reactive microglia from Piry virus–infected animals and uninfected controls showed more complex trees and thicker branches covering a greater tissue volume in the infected animals. Thus, the relative energy cost per monitored unit of tissue volume through vigilant and reactive microglia may impose a metabolic trade-off between the neural (surveillance) and immunological (reactive) functions of these cells. Using an unbiased stereological approach to select microglia for hierarchical cluster and discriminant analyses, we identified a kaleidoscope of microglial morphologies. Given that in nature, form precedes function, our findings are a good starting point for research into integrative approaches for understanding microglial form and function. …”

Reviewer 2 Report

The manuscript of da Silva Creao et al. investigates the morphology of microglial branches in animal infected with an arbovirus and in control mice using advanced bioinformatics and multivariate statistical analyses. Viral encephalitis causes tremendous suffering around the world and research regarding microglial behavior in this disease is an interesting topic. Overall, the manuscript is interesting and timely. However, it needs some clarification and revision upon publication.

Major points

  1. The title states: in limbic encephalitis. However, there is nothing in the introduction nor in the discussion pointing out why the limbic system is of such interest. Please clarify why you chose to investigate the microglia solely in the limbic system. Furthermore, since the manuscript is of interest for virologists, who might not be so familiar with brain anatomy, please clarify CA1/CA2. I can only find the clarification: hippocampal regions in the Materials and Methods.

  1. Another more general remark, the order of the results is not the most logical build up. Please go through the result section carefully and provide pivotal information for understanding first before going into more detail.

  1. The authors state that they are investigating reactive microglia caused by viral encephalitis. However, I cannot find any proof in the manuscript that the brains were indeed infected or that there was virus detectable in the brain. Please provide evidence, for example using histology for viral proteins, that the brains you included were indeed infected. The authors might have provided this in the previous publication, but the reader should not go back to one of the references to see that the brains included in this study are indeed infected. Apart from the observations of the authors, that the microglia have distinct morphology in the infected group, I cannot find evidence, based on conventional techniques, that the included microglia are indeed reactive.

  1. The authors state that they use brain slices from animals included in a previous study. However, while mentioning this, the authors refer to two different references and published studies, namely reference 23 and 34. This is very confusing, what is the original study you are building on?

  1. The authors state: “the iba1 antibody” which one is it, in which concentration was it used, what was the staining protocol? The whole analysis is based on this immunostaining but all information is lacking. They do mention it was published elsewhere but no reference is stated.

  1. Please clarify how you determine whether the entire branches are included in the 70 micron slices. If the branches are not complete within the analyzed slices, this influences all the morphological parameters such as the total and mean branch volume and the size of the tissue volume covered by the microglial trees.

  1. Figure 3B: the SD is very big, are the differences in mean branch volume significant?
  2. Table 4 shows the outcome of two different multiple comparison corrections with similar or even identical outcomes. The inclusion of all these results makes the table very large without additive value. I suggest to only show one in the result section and the other one can go to supplementary to make it easier to read. Furthermore, please include the individual significance level of the morphometric variables in this table.

  1. The authors state that the binary classification of microglia in M1 and M2 phenotypes might be an oversimplification considering their results. In my opinion this is a very strong claim to make based on branch morphology alone. The M1 and M2 states of microglia are not solely based on morphology. Furthermore, the authors do not provide evidence that, based on other M1 or M2 features such as cytokine production, the investigated microglia belong to either of these groups.

  1. In general, the manuscript needs a thorough read through regarding English grammar and consistency in use of abbreviations. In some sentences words are missing and the use of particles is not optimal which hampers fluent reading of the manuscript. Furthermore, in the conclusion it seems like a whole sentence part is missing: These findings may have significant metabloc implications for Using unbiased sterological approach to select microglia for reconstruction….

Always introduce an abbreviation first, then consistently use it afterwards. In materials and Methods, provide company, home base and country.

Minor points

  1. Please clarify how old the mice were exactly. In the first paragraph, it is stated the 10 mice were 2 months old. However, in the section “Animals and infection, it is stated that the animals were 12 weeks of age, which does not correspond to 2 months.

  1. Please check all references in the manuscript. Some references in the discussion are not incorporated in the reference list and are not indicated by numbers. Furthermore, I noticed that some references are in the reference list twice.

  1. Please do not repeat Materials and Methods nor Figure legends in the running text of the result section

  1. we investigated the presence of morphological features” Morphological features are always present, what do you really investigate?

  1. Please reference the Ward´s method

Author Response

Reviewer 2

Comments and Suggestions for Authors

The manuscript of da Silva Creao et al. investigates the morphology of microglial branches in animal infected with an arbovirus and in control mice using advanced bioinformatics and multivariate statistical analyses. Viral encephalitis causes tremendous suffering around the world and research regarding microglial behavior in this disease is an interesting topic. Overall, the manuscript is interesting and timely. However, it needs some clarification and revision upon publication.

Major points

The title states: in limbic encephalitis. However, there is nothing in the introduction nor in the discussion pointing out why the limbic system is of such interest. Please clarify why you chose to investigate the microglia solely in the limbic system. Furthermore, since the manuscript is of interest for virologists, who might not be so familiar with brain anatomy, please clarify CA1/CA2. I can only find the clarification: hippocampal regions in the Materials and Methods.

Reply: As suggested we now added to the Introduction the required paragraphs as follows: “…. In a previous study, we induced Piry viral sublethal encephalitis in an adult albino Swiss mouse model and studied the resulting neuropathological damage and behavioral changes under controlled environmental conditions [23]. We found that Piry neurotropic neuroinvasion through the olfactory epithelium induces encephalitis characterized by inflammation along the limbic areas of the brain, including the olfactory pathways, septum, hippocampus, and amygdala, with rapid activation of microglia. We reconstructed microglia three dimensionally from the CA3 hippocampus field of control in uninfected and Piry virus–infected animals and demonstrated the utility of morphometry and hierarchical cluster analysis for distinguishing between the two conditions [24].

Three fundamental problems require attention before cells can be classified by their morphology in the CNS and other tissues: unambiguous identification with selective markers, definition of readily recognizable boundaries of the sampling area, and application of a random and systematic stereological sample approach to ensure that the reconstructed cell sample within these boundaries is free from bias [25-28]. Computerized microscopy systems allow for 3D reconstruction, recording the position of the soma and branches, tree origin, and process thickness; assigning an order to the traced points; identifying bifurcation and terminal points; differentiating natural terminations from cut ends; and merging tree segments from serial sections [29-34]. Reconstructed cells are then morphologically classified using multivariate statistical analysis of their morphometric features with hierarchical cluster, principal component, and discriminant analyses [35-37].

Here, we revisited the slide collections from our previous study of CA3 [23] and reconstructed microglia from the CA1/CA2 hippocampus field of the same animals, all maintained under standard laboratory conditions. We used a stereological approach for unbiased selection of microglia used in 3D microscopic reconstructions. Using multivariate statistical analysis of morphometric features and distance matrix analysis of both uninfected and infected groups, we searched for morphological changes and found many differential morphotypes between the two groups that may reflect underlying functional changes imposed by homeostatic disruption…”

Another more general remark, the order of the results is not the most logical build up. Please go through the result section carefully and provide pivotal information for understanding first before going into more detail.

Reply: As requested, we went through the results section and provided pivotal information for understanding first before going into more detail as follows: “…

 3.1. Microglial metamorphosis: from surveillance to reactivity to virus encephalitis

“…Immunostaining for Piry virus antigens in the mouse brain after 8 days of nasal instillation of an infectious suspension is illustrated in Figure 1. Virus antigens in cell targets were found in the cytoplasm of infected cells along the olfactory pathways. For further details and pictorial documentation, see [36]. …”

The authors state that they are investigating reactive microglia caused by viral encephalitis. However, I cannot find any proof in the manuscript that the brains were indeed infected or that there was virus detectable in the brain. Please provide evidence, for example using histology for viral proteins, that the brains you included were indeed infected. The authors might have provided this in the previous publication, but the reader should not go back to one of the references to see that the brains included in this study are indeed infected. Apart from the observations of the authors, that the microglia have distinct morphology in the infected group, I cannot find evidence, based on conventional techniques, that the included microglia are indeed reactive.

Reply: As suggested we now included a new Figure (Figure 1) to demonstrate that animals were indeed infected as follows: “…Immunostaining for Piry virus antigens in the mouse brain after 8 days of nasal instillation of an infectious suspension is illustrated in Figure 1. Virus antigens in cell targets were found in the cytoplasm of infected cells along the olfactory pathways. For further details and pictorial documentation, see [36]. …”

Figure 1. Low- and medium-power photomicrographs of a parasagittal section from the brain of a mouse infected with Piry virus, at day 8 post-nasal instillation. A diagram at the corresponding level of a mouse brain stereotaxic atlas [44] can be used to identify the stained neuroanatomical areas with virus antigens (green areas). Immunostained neurons and glial cells are indicated by arrows and arrows head, respectively. VP = ventral pallidum; AAD = anterior amygdaloid area; AAV = anterior amygdaloid area – ventral part; MeAD = medial amygdaloid nucleus, anterior dorsal; MeAV = medial amygdaloid nucleus, anteroventral part; MePD = medial amygdaloid nucleus, posterodorsal part; MePV = medial amygdaloid nucleus, posteroventral part; CA3 = field CA3 of hippocampus;  amygdalohippocampal area; Or = oriens layer of hippocampus; Py = pyramidal cell layer of hippocampus; Rad = Stratum radiatum of the hippocampus; Mol = molecular layer of dentate gyrus; GrDG = granular layer of dentate gyrus; PoDG = Polymorph layer of the dentate gyrus.

The authors state that they use brain slices from animals included in a previous study. However, while mentioning this, the authors refer to two different references and published studies, namely reference 23 and 34. This is very confusing, what is the original study you are building on?

Reply: As suggested we now clarified this issue as follows: “...The infected and control animals used in this work were limited to those from a previous investigation [23], all maintained in standard laboratory cages at 15 m above sea level [38]….”

The authors state: “the iba1 antibody” which one is it, in which concentration was it used, what was the staining protocol? The whole analysis is based on this immunostaining but all information is lacking. They do mention it was published elsewhere but no reference is stated.

Reply: As suggested we now included the requested information as follows: “…For 3D reconstructions, microglia from control and infected animals were immunolabeled with 2 µg/ml of an antibody to IBA-1, an adapter molecule that binds to calcium in macrophages/microglia and is specifically expressed in these cells. It is upregulated during microglia activation [42,43]. A detailed protocol for the immunostaining has been published elsewhere [24]. In brief, the anti-IBA-1 antibody (anti-Iba1, #019-19741; Wako Pure Chemical Industries Ltd., Osaka, Japan) was diluted in PBS (pH 7.2) and used to immunolabel an alternate series of sections to detect microglia and/or macrophages. For immunolabeling, free-floating sections were pre-treated with 0.2 M boric acid, pH 9, at 65–70°C for 60 minutes to improve antigen retrieval, washed in 5% PBS, immersed for 20 minutes in 10% casein (Vector Laboratories, Burlingame, CA, USA), and then incubated with anti-Iba1 (2 µg/ml in PBS, diluted in 0.1 M PBS, pH 7.2¬–7.4), for 3 days at 4°C with gentle and continuous agitation. Washed sections were then incubated overnight with secondary antibody (biotinylated goat anti-rabbit, 1:250 in PBS; Vector Laboratories, Burlingame, CA, USA). Endogenous peroxidases were inactivated by immersion of the sections in 3% H2O2/PBS, then washed in PBS and transferred to a solution of avidin-biotin-peroxidase complex (Vectastain ABC kit; Vector Laboratories, Burlingame, CA, USA) for 1 hour. The sections were washed again before incubation in 0.1 M acetate buffer, pH 6.0, for 3 minutes, and developed in a solution of 0.6 mg/ml diaminobenzidine, 2.5 mg/ml ammonium nickel chloride, and 0.1 mg/ml glucose oxidase. After immunolabeling, all sections were counterstained by cresyl violet. Immunoreacted sections were mounted on gelatinized slides, dried at room temperature, and subsequently dehydrated in alcohols in increasing concentrations (70%, 80%, 90%, and 100%), diaphanized in xylol, and covered with a coverslip using Entellan® (Merck KGaA, Darmstadt, Germany). …”

Please clarify how you determine whether the entire branches are included in the 70 micron slices. If the branches are not complete within the analyzed slices, this influences all the morphological parameters such as the total and mean branch volume and the size of the tissue volume covered by the microglial trees.

 Reply: As suggested we now clarify how we determine that the entire branches are included in 70 µm slices as follows: “…For quantitative analysis, we included only cells with true branch ends within a single section. Cells with branches that seemed to have been cut during sectioning were not included. Thus, all cells with cut end branches extending to the top or bottom of a section under analysis were discarded…”

Figure 3B: the SD is very big, are the differences in mean branch volume significant?

Reply: Yes, the differences in mean branch volume are significant (p values are in Table 3 and Table 4). Please see Colum “Sig.” in Table 3 and Table 4.  

Table 4 shows the outcome of two different multiple comparison corrections with similar or even identical outcomes. The inclusion of all these results makes the table very large without additive value. I suggest to only show one in the result section and the other one can go to supplementary to make it easier to read. Furthermore, please include the individual significance level of the morphometric variables in this table.

Reply: As suggested we adjusted Table 4 limiting the outcomes to Tukey test and the individual significance levels of the morphometric variables are exhibited under “Sig.” columns of Tables 3 and 4.

The authors state that the binary classification of microglia in M1 and M2 phenotypes might be an oversimplification considering their results. In my opinion this is a very strong claim to make based on branch morphology alone. The M1 and M2 states of microglia are not solely based on morphology. Furthermore, the authors do not provide evidence that, based on other M1 or M2 features such as cytokine production, the investigated microglia belong to either of these groups.

Reply: As suggested we delete comments about oversimplification of the binary M1 and M2 classification.

In general, the manuscript needs a thorough read through regarding English grammar and consistency in use of abbreviations. In some sentences words are missing and the use of particles is not optimal which hampers fluent reading of the manuscript.

Reply: We now subjected the whole manuscript to the editorial and proofreading services of San Francisco Edit and we hope English grammar is now correct.

Furthermore, in the conclusion it seems like a whole sentence part is missing: These findings may have significant metabloc implications for Using unbiased sterological approach to select microglia for reconstruction….

Reply: A suggested we changed conclusions as follows: “…In this work, comparison of morphometric measurements of reactive microglia from Piry virus–infected animals and uninfected controls showed more complex trees and thicker branches covering a greater tissue volume in the infected animals. Thus, the relative energy cost per monitored unit of tissue volume through vigilant and reactive microglia may impose a metabolic trade-off between the neural (surveillance) and immunological (reactive) functions of these cells. Using an unbiased stereological approach to select microglia for hierarchical cluster and discriminant analyses, we identified a kaleidoscope of microglial morphologies. Given that in nature, form precedes function, our findings are a good starting point for research into integrative approaches for understanding microglial form and function. …”

Always introduce an abbreviation first, then consistently use it afterwards. In materials and Methods, provide company, home base and country.

Reply: We now consistently introduced abbreviation first and used it afterwards, and in experimental procedures we now provided company, home base and country.

Minor points

Please clarify how old the mice were exactly. In the first paragraph, it is stated the 10 mice were 2 months old. However, in the section “Animals and infection, it is stated that the animals were 12 weeks of age, which does not correspond to 2 months.

Reply: We now clarified this issue in the first paragraph of Experimental Procedures.

Please check all references in the manuscript. Some references in the discussion are not incorporated in the reference list and are not indicated by numbers. Furthermore, I noticed that some references are in the reference list twice.

Reply: As suggested we now checked all references in the manuscript.

Please do not repeat Materials and Methods nor Figure legends in the running text of the result section

Reply: We deleted all sentences that repeat Experimental Procedures or Figure legends in the running of the result section.

“we investigated the presence of morphological features” Morphological features are always present, what do you really investigate?

Reply: As suggested we now improved the sentence: “… Initially, we searched for shared morphometric features among the reconstructed microglia in our sample, within each experimental group. All quantitative morphometric variables with multimodality indexes (MMIs) >0.55 were selected for cluster analysis (Ward’s hierarchical clustering method), which included all animals in each group. Ward’s method relies on the minimum variance criterion to minimize within-cluster variance [46] and has been used to classify microglia [36] and other cells [35,47] by morphometric features. We estimated the MMI based on the asymmetry and kurtosis of our sample for each morphometric variable according to the equation: MMI = [M32 +1] / [M4 + 3 (n - 1)2 / (n – 2) (n – 3)], where M3 is asymmetry, M4 is kurtosis, and n is the sample size. Kurtosis and asymmetry describe the form of data distribution and distinction among unimodal, bimodal, and multimodal curves [35,48]. …”

Please reference the Ward´s method

We now included the reference for Ward’s Method as follows: “…Ward`s Method uses the minimum variance criterion to minimize within-cluster variance (Ward, 1963)and has been used to classify microglia (Yamada & Jinno, 2013) and other cells (Schweitzer & Renehan, 1997; Costa & Velte, 1999) using morphometric features…”

Please see the complete reference on the Reference List

References

Blackman, A.V., Grabuschnig, S., Legenstein, R. & Sjöström, P.J. (2014) A comparison of manual neuronal reconstruction from biocytin histology or 2-photon imaging: morphometry and computer modeling. Front Neuroanat, 8, 65.

Capowski, J.J. (1977) Computer-aided reconstruction of neuron trees from several serial sections. Comput Biomed Res, 10, 617-629.

Costa, L.a.F. & Velte, T.J. (1999) Automatic characterization and classification of ganglion cells from the salamander retina. J Comp Neurol, 404, 33-51.

de Sousa, A.A., Reis, R., Bento-Torres, J., Trevia, N., Lins, N.A.D., Passos, A., Santos, Z., Diniz, J.A.P., Vasconcelos, P.F.D., Cunningham, C., Perry, V.H. & Diniz, C.W.P. (2011) Influence of Enriched Environment on Viral Encephalitis Outcomes: Behavioral and Neuropathological Changes in Albino Swiss Mice. Plos One, 6.

Glaser, E.M. & Vanderloos, H. (1965) A Semi-Automatic Computer-Microscope for the Analysis of Neuronal Morphology. IEEE Trans Biomed Eng, 12, 22-31.

Parekh, R. & Ascoli, G.A. (2013) Neuronal morphology goes digital: a research hub for cellular and system neuroscience. Neuron, 77, 1017-1038.

Schweitzer, L. & Renehan, W.E. (1997) The use of cluster analysis for cell typing. Brain Res Brain Res Protoc, 1, 100-108.

Sheth, C., Ombach, H., Olson, P., Renshaw, P.F. & Kanekar, S. (2018) Increased Anxiety and Anhedonia in Female Rats Following Exposure to Altitude. High Alt Med Biol, 19, 81-90.

Wann, D.F., Woolsey, T.A., Dierker, M.L. & Cowan, W.M. (1973) An on-line digital-computer system for the semiautomatic analysis of Golgi-impregnated neurons. IEEE Trans Biomed Eng, 20, 233-247.

Ward, J. (1963) Hierarchical grouping to optimize an objective function. Journal of the American Statistical Association, 58, 236-244.

Yamada, J. & Jinno, S. (2013) Novel objective classification of reactive microglia following hypoglossal axotomy using hierarchical cluster analysis. J Comp Neurol, 521, 1184-1201.

Zsuppán, F. (1984) A new approach to merging neuronal tree segments traced from serial sections. J Neurosci Methods, 10, 199-204.

This manuscript is a resubmission of an earlier submission. The following is a list of the peer review reports and author responses from that submission.